# Nanoscale Dispersion of Carbon Nanotubes in a Metal Matrix to Boost Thermal and Electrical Conductivity via Facile Ball Milling Techniques

**DOI:** 10.3390/nano13202815

**Published:** 2023-10-23

**Authors:** Bin Li, Lihua Zhou, Bo Wang, Maoshu Yin, Yong Qian, Xianglei Shi, Zhejun Guo, Zhao Han, Nantao Hu, Lijie Sun

**Affiliations:** 1Research Center for Photovoltaics, Shanghai Institute of Space Power-Sources, Shanghai 200245, China; tolb10@163.com (B.L.); zhou_lihua1983@163.com (L.Z.); 15522053272@163.com (B.W.); ymaoshu@163.com (M.Y.); bbqianyong@163.com (Y.Q.); shixianglei99@163.com (X.S.); gzj98762@sjtu.edu.cn (Z.G.); 2Key Laboratory of Thin Film and Microfabrication Technology (Ministry of Education), School of Electronics, Information and Electrical Engineering, Shanghai Jiao Tong University, Dong Chuan Road No. 800, Shanghai 200240, China; zwboss@sjtu.edu.cn

**Keywords:** carbon nanotubes, ball milling, metal composite, electrical, thermal, Young’s modulus

## Abstract

Carbon nanotube (CNT)/metal composites have attracted much attention due to their enhanced electrical and thermal performance. How to achieve the scalable fabrication of composites with efficient dispersion of CNTs to boost their performance remains a challenge for their wide realistic applications. Herein, the nanoscale dispersion of CNTs in the Stannum (Sn) matrix to boost thermal and electrical conductivity via facile ball milling techniques was demonstrated. The results revealed that CNTs were tightly attached to metal Sn, resulting in a much lower resistivity than that of bare Sn. The resistivity of Sn with 1 wt.% and 2 wt.% CNTs was 0.087 mΩ·cm and 0.056 mΩ·cm, respectively. The theoretical calculation showed that there was an electronic state near the Fermi level, suggesting its electrical conductivity had been improved to a certain extent. In addition, the thermal conductivity of Sn with 2 wt.% CNTs was 1.255 W·m^−1^·K^−1^. Moreover, Young’s modulus of the composites with CNTs mass fraction of 10 wt.% had low values (0.933 MPa) under low strain conditions, indicating the composite shows good potential for various applications with different flexible requirements. The good electrical and thermal conductive CNT networks were formed in the metal matrix via facile ball milling techniques. This strategy can provide guidance for designing high-performance metal samples and holds a broad application potential in electronic packaging and other fields.

## 1. Introduction

Ever since the invention of transportation such as airplanes, people have recognized the need for lightweight and high-strength materials [1,2,3,4,5]. The strength and stiffness of these materials have been increased, while their size and mass have been decreased to better meet the requirements of various applications. This has a series of advantages in aircraft, cars, and other manufacturing fields. In order to better meet the requirements of strength and stiffness, some high-performance metal matrix composites have been developed. For such materials, strength and ductility are determined by the matrix, while stiffness is closely related to ceramic or reinforced materials. The low coefficient of expansion and high thermal conductivity of metal-based materials make them ideal candidates for electronic packaging applications as well as automotive and aerospace manufacturing.

Metal tin (Sn), which exhibits excellent collision buffering properties and good electrical and thermal conductivity, has great application potential in those equipment [6]. In recent years, a lot of research has been focused on the strengthening effect of carbon nanotubes (CNTs) in different materials such as polymers, ceramics, and metals [7,8,9,10]. CNTs are a kind of high-performance nanomaterials in which most of the carbon atoms exist in the form of hexagonal crystals. The unique one-dimensional structures can ensure that CNTs connect the micro scale and macro scale [11]. It is found that the electronic properties of multi-walled and single-walled CNTs are basically the same under ideal conditions; in this case, the electrons are transmitted via ballistic mode. CNTs have a dense one-dimensional electron structure that allows them to carry a high current even with little heating, which can greatly improve the electrical and thermal conductivity of composite with tightly interlinked structures. Phonons also propagate easily along CNTs: the measured thermal conductivity of a single CNT at room temperature (3000 W·m^−1^·K^−1^) is greater than that of the base plane of natural diamond and graphite (2000 W·m^−1^·K^−1^) [12].

Metal matrix composites can be prepared by various methods, such as solid phase, liquid phase, semi-solid state, and in situ preparation. Samples prepared by hybridization with other materials, such as ceramic and CNTs with metal as the matrix, are called metal matrix composites. Due to the excellent mechanical, thermal conductivity, and other properties of such materials, metal matrix composites have many applications [13,14]. Compared to bare metal matrix, most metal matrix composites show higher durability, ductility, Young’s modulus, thermal conductivity, etc. [15,16]. By introducing different materials into the metal matrix, it can be applied to different application scopes, such as electronics, aircraft, robots, and other application scopes [17,18]. The high strength of carbon fibers led to their use in missile cone tips, rocket cone exit parts, and heat shields [19,20]. Nanofibers, such as carbon black and CNTs, are introduced into the matrix of metal matrix composites to give them better performance, which are called metal matrix nanocomposites. Materials prepared in this way are more robust and suitable for applications requiring high strength [21,22,23]. Although a great advance has been achieved for metal matrix composites, it is still challenging to achieve facile and scalable fabrication of the metal matrix composites, especially with efficient dispersion of nanomaterials in the matrix, to boost their performance for wide realistic applications.

In this work, the nanoscale dispersion of CNTs in the Sn matrix to boost the thermal and electrical conductivity via facile ball milling techniques was reported. The CNT-Sn composite material was obtained using sintering processes under certain conditions. We found that the resistivity of the prepared CNT-Sn was lower than that of bare CNTs, and the resistivity of some samples was lower than that of bare metal Sn. In the test of stress and strain, Young’s modulus of the material is lower, which can well buffer the potential energy generated by collision.

## 2. Experimental

### 2.1. Preparation and Carboxylation of CNTs

The CNTs were first prepared by chemical vapor deposition (CVD). The nickel catalyst was deposited on the surface of the silicon wafer via direct current sputtering to form a layer of nanometer-thick nickel oxide film. Then, the carrier sheet was put into a vacuum furnace with hydrogen at 800 °C for 15 min to obtain fine nickel particles on the carrier surface. The mixed gas with a volume ratio (2:1) of acetylene to nitrogen was purged into the furnace. Then, CNTs were obtained at high temperatures.

The CNTs were modified in aqua regia at room temperature for three hours to make the carboxylated surface of the CNTs. Then, the CNTs were centrifuged in ethanol and cleaned five times (centrifugation speed: 5000 rpm) to obtain the carboxylated CNTs.

### 2.2. Preparation of CNTs-Sn Composites

Firstly, a certain amount of Sn powder and a certain amount of carbon nanotube powder were weighed, and the mass ratio of CNTs was 2, 5, 10, and 20 wt.%, respectively. After the carboxylated CNTs and Sn powder were mixed evenly, they were put into the micro-planetary ball mill, and the ratio of ball to material was about 5:1. The ball milling started at 400 rpm and kept operating for 6 h.

At the same time, for the sample with 2 wt.% CNTs, we conducted control experiments with ball milling time of 6 h, 4 h, and 2 h, respectively, to test the influence of ball milling time on the properties of the composite. After the ball milling was completed, the samples in the ball milling tank were taken out, centrifuged, and cleaned five times (centrifugation speed: 10,000 rpm). The CNT-Sn composites were obtained by freeze-drying for 12 h.

### 2.3. Characterization of CNT-Sn Composites

The morphology characteristics of the CNT-Sn composites obtained by centrifugation were observed using a field emission scanning electron microscope (FESEM, Zeiss Ultra 55, 5 kV, Jena, Germany), and the elemental composition and proportion of CNT-Sn composites were measured by energy spectrum (EDS, Zeiss Ultra 55, 5 kV, Jena, Germany). A Reflex Raman spectrometer from Renishaw was used to analyze the related molecular structures of the CNT-Sn composites. X-ray diffraction (XRD, D8, Bruker, Mannheim, Germany) spectra of the CNT-Sn composites were obtained. Fourier infrared spectra (FTIR, VERTEX 70, Bruker Optik GmbH, Ettlingen, Germary) of the CNT-Sn composites were measured using an infrared spectrometer. The X-ray photoelectron spectroscopy (XPS, AXIS UltraDLD, Kratos, Hadano, Japan) spectra of the CNT-Sn composites were tested using an X-ray electron spectrometer.

### 2.4. CNTs-Sn Composite Laminating and Sintering

The obtained CNT-Sn powder was put into the powder resistance tester for tablet pressing, and the resistance changes in the CNT-Sn powder under different pressures were measured at the same time. The final pressure of the powder resistance meter was 30 MPa. The sample was taken out at 30 MPa for 10 s. Then, it was placed in the furnace under nitrogen flow for sintering for 6 h. The sintering temperature was 200 °C, and the corresponding speed was 5 °C per minute in the heating process.

### 2.5. Performance Test of the CNT-Sn Sintered Composite Sample

The resistivity and square resistance of the CNT-Sn sintered composite samples were measured using a four-probe powder resistance meter. A dynamic thermomechanical analyzer (DMA 8000, PerkinElmer, Waltham, MA, USA) was used to measure the stress–strain curves of the CNT-Sn sintered composite samples.

### 2.6. Theoretical Calculation Methods

We employed the Vienna Ab initio Simulation Package (VASP) [24,25] to perform all the density functional theory (DFT) calculations within the generalized gradient approximation (GGA) using the Perdew–Burke–Ernzerhof (PBE) formulation [26]. The projected augmented wave (PAW) potential [27,28] was selected to describe the ionic cores and take valence electrons into account using a plane wave basis set with a kinetic energy cut-off of 520 eV. The density of the kmeshs grids for Brillouin zone sampling was set as 0.04 × 2π/Å. Partial occupancies of the Kohn–Sham orbitals were allowed using the Gaussian smearing method and a width of 0.05 eV. Electronic energy was considered self-consistent when the energy change was smaller than 10^−6^ eV. Geometry optimization was considered convergent when the force on each atom was smaller than 0.02 eV/Å.

## 3. Results and Discussions

As displayed schematically in Figure 1, the CNT-Sn composite was prepared by ball milling techniques. The method is not only simple and low-cost but also easily scaled up.

Appendix A shows the SEM images of the CNTs. The CNTs show a curly shape with a diameter of about 10–15 nm. Appendix A shows the SEM of Sn powders. The diameter of Sn particles is between 1 and 10 μm. In addition, the surface of Sn particles is relatively smooth, and the structure of Sn itself is very stable.

As shown in Figure 2a–c, with the increase in CNT content, the curled shape of CNTs becomes more and more obvious. The CNTs are wrapped around the Sn surface to form the interconnected network structure, which can further promote the electrical conductivity of the composite structure. That is because metal Sn powder is broken after a ball mill, and no longer smooth surface can be observed at the same time. At high magnification, CNTs can be found to be intimately attached to metal Sn, which also proves that CNTs and Sn are successfully compounded. The insert of Figure 2 shows the EDS analysis diagram of CNT-Sn composite material with 10 wt.% CNTs. It can be seen that the mass fraction of the C element is about 26.8%. The EDS analysis diagram also sees the existence of oxygen; this may be attributed to a certain quantity of heat in the process of the ball mill, resulting in an oxide layer at the surfaces.

Figure 2d shows the XRD spectra of CNT-Sn composite powders with different CNT mass fractions after ball milling. The two peaks within 30~35 degrees correspond to the (200) and (101) crystal planes of elemental Sn. The two peaks between 40 and 50 degrees represent the (220) and (221) crystal planes of metallic Sn, and the peaks around 55 degrees represent the (301) crystal planes of metallic Sn. In addition, the three peaks between 60 and 70 degrees represent the (112), (400), and (321) crystal faces of Sn metal, and the two peaks between 70 and 75 degrees represent the (420) and (421) crystal faces, respectively. The peak located at 80 degrees represents the (312) crystal face of metal Sn [29]. It can be observed that no matter what the mass ratio of CNTs is, the characteristic peak of metal Sn in the composite material can be obviously observed. And when the proportion of metal Sn changes constantly, the peak intensity also changes with the different proportions of metal Sn, which is the same as the proportion of composite material in our experiment. At the same time, the existence of oxygen element is also seen in the EDS analysis diagram, which may be due to the reason that metal Sn we select is granular metal Sn with the activeness of metal Sn itself. It is a protective layer of Sn oxide attached to the surface. The peak between 25° and 30° represents the characteristic peak of CNTs. Due to the low content of CNTs, the characteristic peak of CNTs is weak, but it can also be seen that the intensity of the characteristic peak of CNTs changes with the proportion of CNTs in the sample. According to XRD patterns, CNT-Sn composites were successfully compounded, and the characteristic peak intensity of CNTs and metal Sn was similar to their mass ratios [30].

Figure 2e shows the Fourier transform infrared spectra of CNTs-Sn composite materials with the CNTs mass fraction ratio of 10, 5, and 2 wt.%, respectively. All three composite materials with different proportions have similar infrared spectra. The diffraction peak at 3400 cm^−1^ indicates the existence of an O-H bond, which may be due to the adsorbed water molecules or hydroxyl groups at the CNT surfaces. The peak in the region of 2800 to 3000 cm^−1^ reflects the symmetrical stretching vibration of C-H. Furthermore, the peak at 1647 cm^−1^ corresponds to the stretching vibration of C=C, and the peak between 1250 and 1500 cm^−1^ indicates the oscillating vibration mode of C-H bond and O-H bond. In addition, the two peaks between 1000 and 1250 cm^−1^ indicate the stretching vibration pattern of the C-O bond [31]. Figure 2f shows the Raman spectra of the CNT and CNT-Sn composites with different CNT mass fractions. There are two obvious Raman scattering bands on both sides of 1500 cm^−1^. Further analysis shows that the corresponding D-peak is located at 1350 cm^−1^, which also indicates that there are certain defects in the carbon nanotube, and the G-peak is located at 1580 cm^−1^. Accordingly, it shows the degree of order of the sample [32]; the existence of the two peaks shows the presence of CNTs in the sample, revealing little structural damage after the ball milling and sintering process. The peak intensity in the Raman spectra can partly be attributed to the content changes in the Sn-CNT composites. With the decrease in the content of CNTs, the peak strength gradually weakened, which was the same as the ratio in the actual sample preparation process. The analysis of the reason for the change showed that this was closely related to the influence of CNTs on the peak strength.

Figure 3a is the XPS full spectrum of the CNT-Sn composites with a 5% mass fraction of CNTs. The main response spectra and valence state of main elements distribution of CNTs-Sn composites after ball milling can be seen in the full spectrum of XPS spectra. It shows that carbon and a small amount of oxygen element. Figure 3b,c shows the narrow spectra of several energy levels of C1s and O1s of CNT-Sn composite materials. In the high-resolution narrow spectrum of C1s, we can observe several major photoemission peak positions after peak splitting. The peak at 284.8, 285.7, 288.8, and 290.9 eV corresponds to sp^2^, sp^3^ hybridized states, O-C=C, and C=O bonds, respectively [33]. The corresponding characteristics in the high-resolution narrow spectrum of C1s are basically consistent with those in the infrared spectrum. Additionally, in the high-resolution narrow spectrum of O1s, the peak at 531.5, 532.5, and 532.7 eV corresponds to the C=O, O-H, and C-O bond, respectively. The key position of the high-resolution narrow spectrum peak of O1s is also basically the same as the key position in the Fourier infrared spectrum. Meanwhile, the key position corresponding to the peak of the high-resolution atlas of C1s and O1s is also matched. Figure 3c shows the high-resolution narrow spectrum of Sn3d of CNTs-Sn composite material with a 5% mass fraction of CNTs. The main peak is the 3d_5/2_ and 3d_3/2_ spectrum of Sn, respectively. The peak at Sn3d_5/2_ can be divided into two independent signals, among which the peak at 485.64 eV represents the peak of bare metal Sn, while the peak at 487.2 eV represents the peak of positive tetravalent Sn [34,35]. The peak intensity of bare metal Sn is much higher than that of positive tetravalent Sn, which proves that most of the composite samples are simple metal Sn. The oxidized Sn is observed because the Sn itself easily forms an oxide layer on the surface due to its activity. Moreover, the process of ball milling may produce a certain high temperature, leading to the partial oxidation of the Sn in the sample. However, there is no bond between the oxygen element and Sn element in the high-resolution narrow spectrum of O1s, which may be due to (1) The content of Sn oxide is lower than that of metal Sn. (2) The peak of Sn oxide coincides with the peak of the C=O key position. In general, corresponding to the high-resolution narrow spectra of O1s, C1s, and Sn3d in CNT-Sn composites are basically consistent, which can also overlap with the valence bonds corresponding to FTIR spectra [36].

Figure 4a shows the resistivity curve of bare metal Sn powder with the pressure change. We can see the pure resistivity of metal powder also decreases as the pressure gradually becomes smaller. This also proves that the resistivity of metal Sn powder can be significantly reduced by the introduction of a small amount of CNTs. Comparative analysis shows that the resistivity of the composite is at the minimum level when the incorporation ratio is 2 wt.%. As can be seen from Figure 4b, when the incorporation ratio of CNTs changes, there is a close relationship between the resistivity and pressure of the corresponding CNT-Sn sample. It can be seen that the resistivity of all proportions of composites decreases gradually with the increase in pressure. The resistivity decreases rapidly at the beginning of the curve, while the value decreases gradually with the increase in pressure up to 30 MPa. It can be known that the powder composite material is loose under low pressure when a little pressure can make their resistivity drop a lot. However, when the pressure is high, the resistivity drop is not obvious when the pressure is applied. Among all the curves of the resistivity changing with pressure, the resistivity curve of the composite with 5 wt.% CNTs decreased the most and reached the minimum value at 30 MPa. For the two samples with the CNTs mass fraction of 10 and 2 wt.%, the resistivity changes slightly with the pressure. The powders with the two proportions of composite samples have a certain density when the pressure is low. At the beginning of the curve, the resistivity of all samples exceeds that of bare CNTs. When the pressure reaches 30 MPa, CNT-Sn composites with different mass fractions of CNTs are much lower than the resistivity of metal Sn powder (26.6 mΩ·cm).

The first principle calculation was introduced to further determine the change in the electronic state. The binding energy of Sn/CNTs was calculated using the following equation:$$E_{b}=E_{tot}+E_{Sn}-E_{CNT}$$
where *E_b_* is the binding energy of Sn/CNTs, *E_tot_* is the total energy of the Sn/CNTs interface, and *E_Sn_* and *E_CNTs_* are the energy of the slab mode of the Sn (111) surface and CNTs, respectively.

Figure 5a shows the structure of the Sn/CNT interface after structural optimization, and the binding energy of Sn/CNTs is −1.70 eV, indicating that the CNTs can spontaneously adsorb on the surface of Sn. In order to study the charge transfer before and after the Sn/CNT interface combination, we calculated the charge density difference of Sn/CNTs, as shown in Figure 5b,c. The yellow region is the area of increased charge density, and the blue region is the area of reduced charge density. Moreover, the charge density on both sides of Sn and CNTs is uniformly reduced to a certain extent and transferred to the interlayer region, reflecting a specific covalent bond characteristic. Furthermore, the density of states before and after combining the CNTs and Sn surfaces was calculated, as shown in Figure 5c. The electronic density of states of pure CNTs has a band gap near the Fermi level, reflecting the semiconductors’ characteristics. However, after the Sn combination, the electron state density moves significantly to the deep energy level, indicating that the CNTs become more stable after binding with Sn. There is an electronic state near the Fermi level, indicating its conductivity has improved to a certain extent.

Figure 6a shows the stress–strain curves of sintered CNT-Sn composites. We can observe that the stress increases with the change in strain. The strain curve of the sample is relatively flat between 0% and 0.8%. The strain curve of the sample gradually becomes steeper with the increase in strain displacement when the strain displacement is between 0.8% and 1.8%. In other words, as the strain displacement increases, Young’s modulus of the sample also gradually increases in Figure 6b. When the strain displacement is small, among the three samples with different CNT ratios, the Young’s modulus of the sample with CNT mass fractions of 10 wt.% is the lowest (0.933 MPa). Also, the Young’s modulus of the samples with 5 and 2 wt.% CNTs is 1.77 and 2.09 MPa, respectively. The introduction of loose CNTs has little effect on the electrical conductivity of Sn, resulting in a small Young’s modulus although the addition content of CNTs is as high as 10 wt.% [37]. The Young’s modulus of composite also increases with the increase in the ratio of CNTs when the strain is large due to the conductive network formed by tightly stacked CNTs.

Under the condition of different ball milling times, the stress and strain curve of composite with an incorporation ratio of 5 wt.% after sintering is shown in Appendix A. When the strain ratio is small, there is a positive correlation between ball milling time and stress- strain curve. The Young’s modulus of the sample gradually increases with the increase in ball milling time. With the increase in ball milling sample preparation time, the conductive network between CNTs and metal Sn compound became closer, leading to the improvement in the stress transfer efficiency. Moreover, the Young’s modulus of the sample increases gradually, as shown in Appendix A. According to the solid conduction theory, the metal is the electron donor in the composite, but the mobility of the metal itself is low. In contrast, CNTs have relatively high mobility and current-carrying capacity. The composite of CNTs and metal Sn fabricated by ball milling has lower resistivity compared with bare CNTs and metal Sn power as a result of the effective synergistic effect. This proves that the CNT-Sn composite material has a good cooperative transmission network due to a good electron donor of metal and a high mobility of CNTs [38]. Therefore, such composite material has a relatively low resistivity and thus has a good electron transmission capacity.

Figure 7a,b shows the resistivity of CNT-Sn composite materials before and after sintering. Comparative analysis shows that the resistivity of all samples decreases to a certain extent after sintering due to the better contact between CNTs and metal Sn with a network of interlocking synergies. At the same time, the resistivity of the sample is also significantly reduced after the metal Sn sheet pressing and sintering. At the sintering temperature of 200 °C, the metal Sn powder is in a micro-melting state and has a better combination compared with the ordinary sheet pressing [39]. Figure 8 shows the resistivity diagram of sintered CNT-Sn composites with CNT mass fractions of 5 wt.% under different ball milling times. The resistivity increases gradually as the ball milling time decreases. With the increase in ball milling time, the composite effect between CNTs and metal Sn is better, resulting in a better synergistic interaction network between CNTs and metal Sn. Figure 9 shows the change curve of thermal conductivity of CNT-Sn at different temperatures with 2 wt.% CNTs. It can be seen that the thermal conductivity of CNTs-Sn can reach the value of 1.255 W·m^−1^·K^−1^. The thermal conductivity of the CNT-Sn was mainly determined by the addition of CNTs, suggesting that the CNT interface has a better transmission efficiency and apparently benefits the thermal conduction. Therefore, the proper amount of CNTs can not only increase the conductivity of the composite but also have better thermal conductivity.

## 4. Conclusions

In conclusion, the nanoscale dispersion of carbon nanotubes in the Sn matrix to boost thermal and electrical conductivity via facile ball milling techniques was demonstrated. The CNT-Sn composites produce a good collaborative interaction network after a tablet pressing and sintering process. The metal Sn as an electron donor with the electron transport of CNTs greatly improves the electron migration efficiency in the composite material, which can be obviously reflected in the change in the resistivity of the sample. Moreover, the resistivity of all samples is lower than that of CNTs prepared under the same conditions. In addition, when the mass fraction of CNTs is 1 wt.% and 2 wt.%, the resistivity of CNT-Sn is lower than that of bare Sn calcined samples due to the excellent conductivity of CNTs networks. The resistivity of CNT-Sn (1 wt.% and 2 wt.%) is 0.087 and 0.056 mΩ·cm, respectively. The theoretical calculation showed that the electron state density moved to the deep energy level, indicating that the CNTs became more stable after binding with Sn. In the stress–strain curve test, Young’s modulus of the sample decreases gradually with the increase in the mass fraction of CNTs in the sample. These composites with excellent thermal and electrical conductivity have a good application prospect in the field of electronic packaging and 3D printing materials.

## Figures and Tables

**Figure 1 nanomaterials-13-02815-f001:**
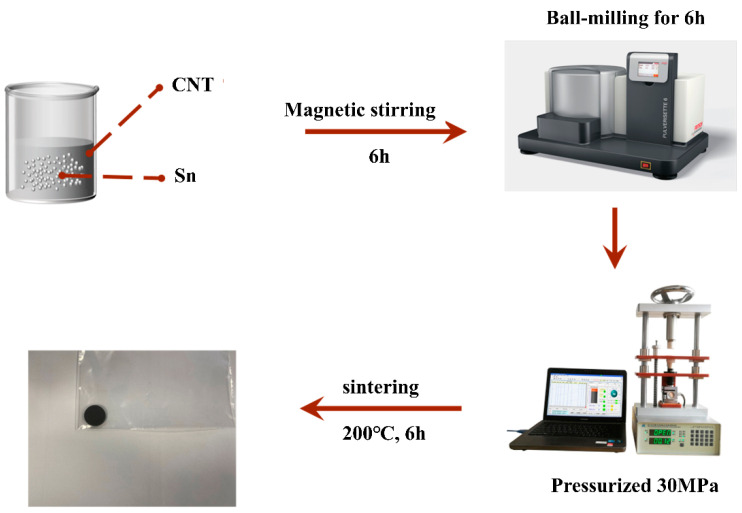
Flow chart of CNT-Sn composite prepared by ball milling.

**Figure 2 nanomaterials-13-02815-f002:**
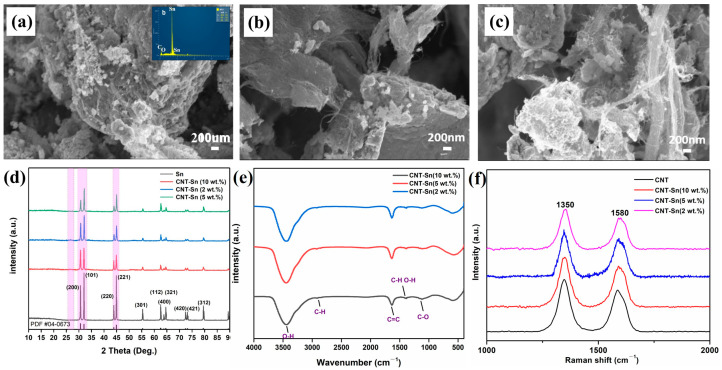
(**a**–**c**) SEM of CNT-Sn composites with mass fractions of 10%, 5%, and 2%, respectively, before sintering. The insert is the EDS of CNT-Sn. (**d**) XRD patterns of CNT-Sn composites with different CNTs mass fractions. (**e**) FTIR spectra of CNT-Sn composites with different CNT mass fractions. (**f**) Raman spectra of CNT-Sn composites with different CNT mass fractions.

**Figure 3 nanomaterials-13-02815-f003:**
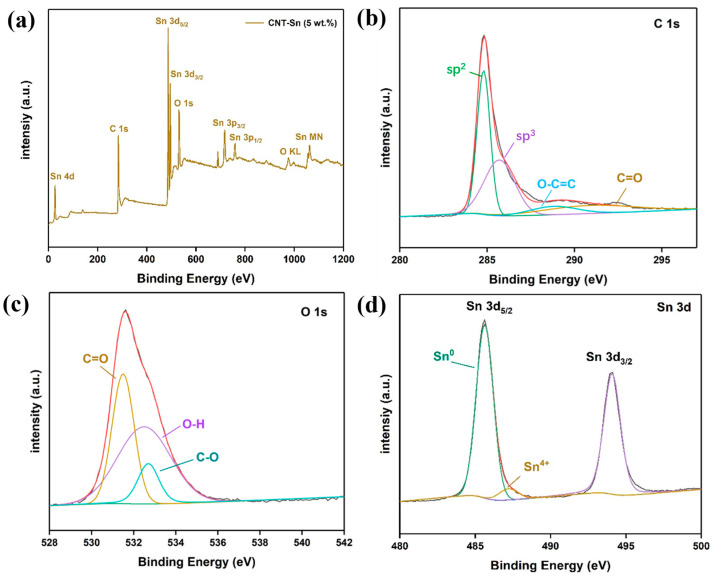
(**a**) XPS full spectrum of 5% CNT-Sn composites; (**b**–**d**) the narrow spectrum of C1s, O1s, and Sn3d in CNT-Sn composites with 5% CNTs.

**Figure 4 nanomaterials-13-02815-f004:**
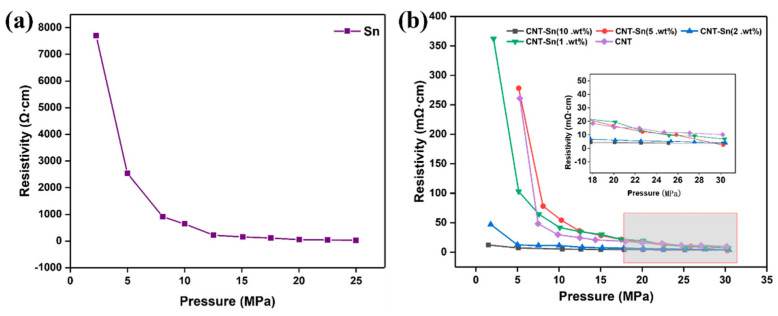
(**a**) Resistivity curve of pure metal Sn powder with pressure. (**b**) Pressure dependence curve of conductivity of CNT-Sn with different CNT mass fraction.

**Figure 5 nanomaterials-13-02815-f005:**
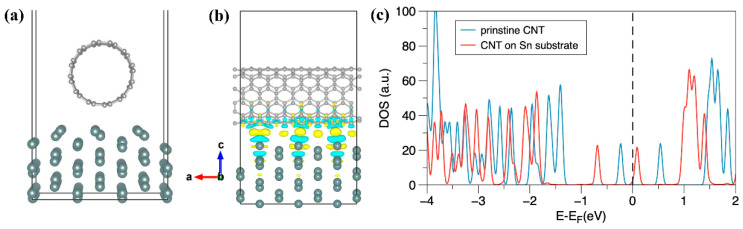
(**a**) The optimized Sn/CNT interface structure model; (**b**) the interface charge density distribution model diagram of Sn/CNTs; (**c**) the density of states before and after combining the CNT and Sn surfaces.

**Figure 6 nanomaterials-13-02815-f006:**
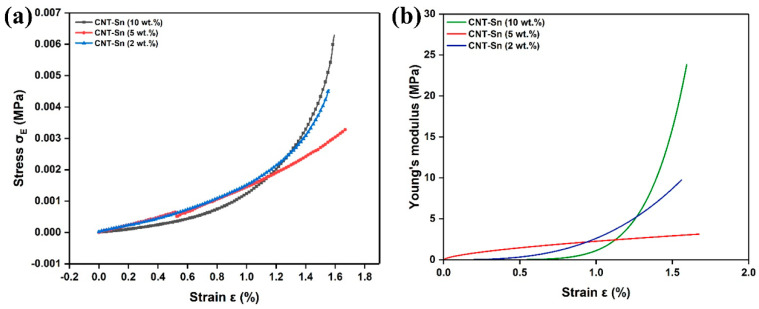
(**a**,**b**) Stress-strain and Young’s modulus curves of sintered CNT-Sn composites with different proportions of CNTs mass fraction, respectively.

**Figure 7 nanomaterials-13-02815-f007:**
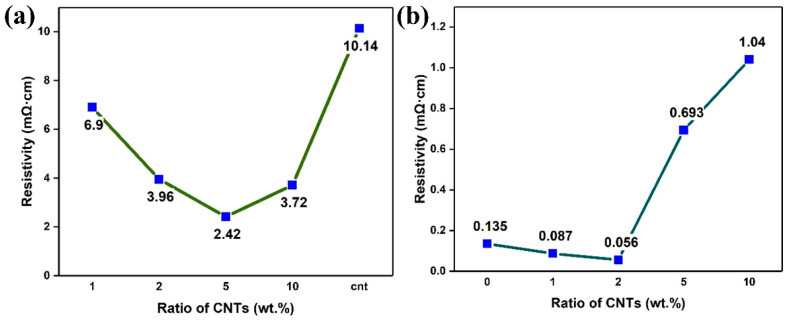
(**a**) Resistivity of CNT-Sn composites with different CNT mass fraction and bare CNT powder before sintering; (**b**) resistivity of CNT-Sn composite with different CNT mass fraction and bare tin after sintering.

**Figure 8 nanomaterials-13-02815-f008:**
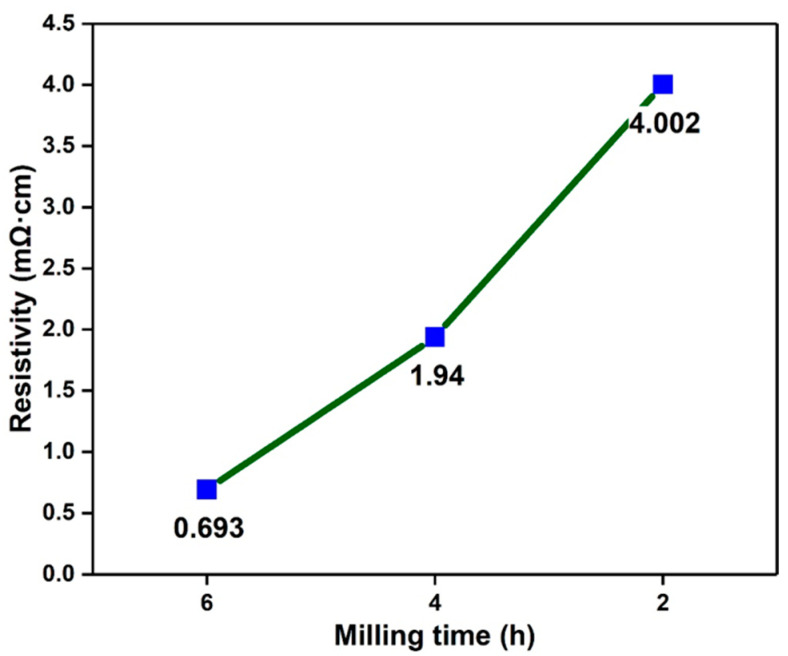
Resistivity of CNT-Sn composite with 5 wt.% CNT after different ball milling times.

**Figure 9 nanomaterials-13-02815-f009:**
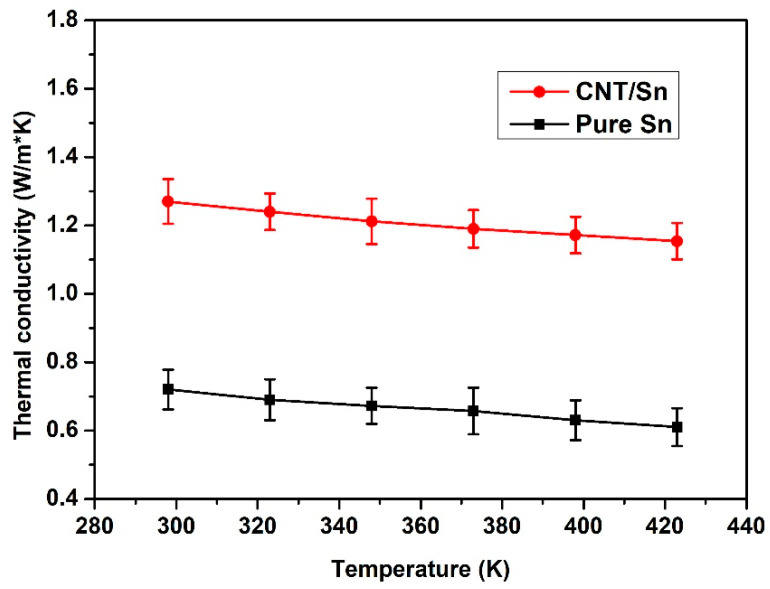
Thermal conductivity curves of CNT-Sn and pure Sn at different temperatures with 2 wt.% CNTs.

## Data Availability

Not applicable.

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
