# Peer review of "Nanoscale Dispersion of Carbon Nanotubes in a Metal Matrix to Boost Thermal and Electrical Conductivity via Facile Ball Milling Techniques"

_nanomaterials, 2023, doi:10.3390/nano13202815_

Round 1
Reviewer 1 Report
This paper reports the synthesis of Sn composites with carbon nanotubes, showing the various performance enhancements such as conductivity, thermal conductivity, and mechanical performance. This paper suggests very interesting experimental results of the effect of carbon nanotubes to obtain enhancement of performances of Sn, which indicates the development of various potential applications using Sn composites. My comments are listed below.
1) There needs to be detailed information about used carbon nanotubes, such as how to prepare.
2) I recommend showing the Raman spectra of pristine carbon nanotube to reveal structural damage by the ball mill and sintering process.
3) The motivation of this paper needs to be clarified. It is better to include its explanation and reason in the introduction, such as why the authors selected Sn and tried to improve its performance by adding carbon nanotubes.
Author Response
Dear Editor and Reviewers:
Thank you for your letter and for the reviewers’ comments concerning our manuscript entitled “Nanoscale dispersion of carbon nanotubes in metal matrix to boost thermal and electrical conductivity via facile ball milling techniques” (ID: nanomaterials-2640870). Those comments are all valuable and very helpful for revising and improving our paper, as well as the important guiding significance to our researches. We have studied comments carefully and have made point-by-point responses to the reviewers. Revised portion are highlight marked in our revised manuscript. The main corrections in our paper and the responds to the reviewer’s comments are as following:
Our point-by-point responses as follows:
Reviewers’ comments and responses:
Reviewer: 1
- There needs to be detailed information about used carbon nanotubes, such as how to prepare.
Response: Thanks very much for the referee’s suggestions. We have perfected the experimental description to make the preparation process of CNT as detailed as possible. The CNTs were prepared by chemical vapor deposition (CVD) method. The changes have been marked in the revised version in the experimental section of “2.1. Preparation and carboxylation of CNTs” (Line 1-6, Page 2, fourth paragraph).
“The CNTs were firstly prepared by chemical vapor deposition (CVD) method. The The nickel catalyst is deposited on the surface of the silicon wafer through direct current sputtering to form a layer of nanometer thick nickel oxide film. Then the carrier sheet is put into a vacuum furnace with hydrogen at 800℃ for 15 minutes to obtain fine nickel particles on the carrier surface. The mixed gas with a volume ratio (2:1) of acetylene to nitrogen is passed into the furnace. Then CNTs were obtained at high temperature.”
- I recommend showing the Raman spectra of pristine carbon nanotube to reveal structural damage by the ball mill and sintering process.
Response: Thanks very much for the referee’s suggestions. The Raman spectra of pristine carbon nanotube had been added in the results and discussions section of Revised Manuscript (Line 18, Page 4).
- The motivation of this paper needs to be clarified. It is better to include its explanation and reason in the introduction, such as why the authors selected Sn and tried to improve its performance by adding carbon nanotubes.
Response: Thanks very much for raising such significant questions. We have perfected the description of motivation of this paper in the introduction to explain why the authors selected Sn and tried to improve its performance by adding carbon nanotubes. Metal tin (Sn), which has excellent collision buffering properties, good electrical and thermal conductivity, has great application potential in electronic packaging applications as well as automotive and aerospace manufacturing. Moreover,CNTs have a dense one-dimensional electron structure that allows them to carry a high current even with little heating, which can greatly improve the electrical and thermal conductivity of composite with tightly interlinked structures. The changes have been marked in the revised version in the introduction section (Line 8-14, Page 1 and Line 2-5, Page 2).
“The low coefficient of expansion and high thermal conductivity of metal-based materials make them ideal candidates for electronic packaging applications as well as automotive and aerospace manufacturing.
Metal tin (Sn), which has excellent collision buffering properties, good electrical and thermal conductivity, has great application potential in those equipment[6]. In recent years, a lot of research has been done on the strengthening effect of carbon nanotubes (CNTs) in different materials such as polymers, ceramics and metals [7-10].”
“CNTs have a dense one-dimensional electron structure that allows them to carry a high current even with little heating, which can greatly improve the electrical and thermal conductivity of composite with tightly interlinked structures.”

Reviewer 2 Report
In the submited manuscript „Nanoscale dispersion of carbon nanotubes in metal matrix to boost thermal and electrical conductivity via facile ball milling techniques” authors show that introduction of CNTs into the stannum matrix by ball milling can affect both the mechanical as well as electrical and thermal properties of the composite. The presented results are an interesting contribution to the knowledge of metal-carbon nanotube composites, and I recommend the paper for publication in Nanomaterials. However, before accepting the paper for publication, the authors should thoroughly improve the English.
Substantive comments:
Superconductivity phenomena mentioned by the authors in the second paragraph of Introduction refers to the transition metal dichalcogenides [see Ref.12].
In section 2.6, 10-6 eV should be changed into 10 to the power of -6.
In section 3, page 5, second paragraph, discussion of the Raman spectra, “diffraction peaks on both sides of 1500 cm-1” should be changed to “Raman scattering bands on both sides of 1500 cm-1”.
The next paragraph in the same page, when the XPS spectra are discussed, authors again confuse photoemission peaks with diffraction peaks. In addition, the peaks at 284.8 and 285.7 eV obtained from the deconvolution of C1s line were erroneously assigned. It should be assigned to sp2 and sp3, respectively. See, f.ex.: M. Rybachuk and J.M. Bell, Electronic states of trans-polyacetylene, poly(p-phenylene vinylene) and sphybridised carbon species in amorphous hydrogenated carbon probed by resonant Raman scattering, Carbon 47(10):2481-2490, 2009, DOI: 10.1016/j.carbon.2009.04.049.
I also suggest moving the Figures S5 and S6 from the Supplementary Materials to the main text.
Before accepting the paper for publication, the authors should thoroughly improve the English.
Author Response
Dear Editor and Reviewers:
Thank you for your letter and for the reviewers’ comments concerning our manuscript entitled “Nanoscale dispersion of carbon nanotubes in metal matrix to boost thermal and electrical conductivity via facile ball milling techniques” (ID: nanomaterials-2640870). Those comments are all valuable and very helpful for revising and improving our paper, as well as the important guiding significance to our researches. We have studied comments carefully and have made point-by-point responses to the reviewers. Revised portion are highlight marked in our revised manuscript. The main corrections in our paper and the responds to the reviewer’s comments are as following:
Our point-by-point responses as follows:
Reviewers’ comments and responses:
Reviewer: 2
- Superconductivity phenomena mentioned by the authors in the second paragraph of Introduction refers to the transition metal dichalcogenides [see Ref.12].
Response: Thanks very much for the referee’s suggestions. We have deleted the inappropriate description and the cited reference about the superconductivity phenomena mentioned in the second paragraph of Introduction.
- In section 2.6, 10-6 eV should be changed into 10 to the power of -6.
Response: Thanks very much for the reviewer’s suggestions. We have revised the formatting error.
- In section 3, page 5, second paragraph, discussion of the Raman spectra, “diffraction peaks on both sides of 1500 cm-1” should be changed to “Raman scattering bands on both sides of 1500 cm-1”.
Response: Thanks very much for the referee’s suggestions. We have revised the inappropriate description about the Raman scattering bands (Line 11, Page 5, second paragraph).
“There are two obvious Raman scattering bands on both sides of 1500 cm-1”
- The next paragraph in the same page, when the XPS spectra are discussed, authors again confuse photoemission peaks with diffraction peaks. In addition, the peaks at 284.8 and 285.7 eV obtained from the deconvolution of C1s line were erroneously assigned. It should be assigned to sp2 and sp3, respectively. See, f.ex.: M. Rybachuk and J.M. Bell, Electronic states of trans-polyacetylene, poly(p-phenylene vinylene) and sphybridised carbon species in amorphous hydrogenated carbon probed by resonant Raman scattering, Carbon 47(10):2481-2490, 2009, DOI: 10.1016/j.carbon.2009.04.049.
Response: Thanks very much for the referee’s suggestions. We have revised the inappropriate description about the photoemission peaks of C1s (Line 7-8, Page 5, third paragraph). Moreover, we have revised the description of the peaks at 284.8 and 285.7 eV obtained from the deconvolution of C1s line and added the reference (Line 8-9, Page 5, third paragraph).
“In the high-resolution narrow spectrum of C1s, we can observe several major photoemission peak positions after peak splitting.”
“The peak at 284.8, 285.7, 288.8 and 290.9 eV corresponds to sp2, sp3 hybridized states and O-C=C, C=O bond, respectively [33].”
- I also suggest moving the Figures S5 and S6 from the Supplementary Materials to the main text.
Response: Thanks very much for the referee’s suggestions. We have moved the the Figures S5 and S6 from the Supplementary Materials to the main text as shown in Figure 7-8 (Page 9).
- Before accepting the paper for publication, the authors should thoroughly improve the English.
Response: Thanks very much for the referee’s suggestions. We have reread and revised the typographical errors in the whole manuscript with the assistance from my colleagues. The changes have been marked in the revised version.

Round 2
Reviewer 1 Report
The revised manuscript shows appropriate significance for Nanomaterials; therefore, I recommend the publication of this paper in the present form.
Reviewer 2 Report
The manuscript has been sufficiently improved and can be accepted for publication in Nanomaterials in the present form.